# GAN-based Symmetric Embedding Costs Adjustment for Enhancing Image Steganographic Security

## ABSTRACT

Designing embedding costs is pivotal in modern image steganography. Many studies have shown adjusting symmetric embedding costs to asymmetric ones can enhance steganographic security. However, most existing methods heavily depend on manually defined parameters or rules, limiting security performance improvements. To overcome this limitation, we introduce an advanced GAN-based framework that transitions symmetric costs to asymmetric ones without the need for the manual intervention seen in existing approaches, such as the detailed specification of cost modulation directions and magnitudes. In our framework, we firstly achieve symmetric costs for a cover image, which is randomly split into two sub-images, with part of the secret information embedded into one. Subsequently, we design a GAN model to adjust the embedding costs of the second sub-image to asymmetric, facilitating the secure embedding of the remaining secret information. To support our phased embedding approach, our GAN's discriminator incorporates two steganalyers with different tasks: distinguishing the generator's final output, i.e., the stego image, from both the input cover image and the partially embedded stego image, providing diverse guidance to the generator. In addition, we introduce a simple yet effective update strategy to ensure a stable training process. Comprehensive experiments demonstrate that our method significantly enhances security over existing symmetric steganography techniques, achieving state-of-the-art levels compared to other methods focused on embedding costs adjustments. Additionally, detailed ablation studies validate our approach's effectiveness.

## CCS CONCEPTS

• **Information systems** → *Multimedia information systems*; • **Security and privacy** → *Security services*;

## KEYWORDS

Image Steganography, Generative Adversarial Networks, Embedding Costs Adjustment, Steganalysis

**ACM Reference Format:**
Anonymous authors. 2024. GAN-based Symmetric Embedding Costs Adjustment for Enhancing Image Steganographic Security. In *Proceedings of the 32th ACM International Conference on Multimedia (MM '24), October 28-November 1, 2024, Melbourne, Australia.* ACM, New York, NY, USA, 12 pages. https://doi.org/XXXXXXX.XXXXXXX

## 1 INTRODUCTION

Image steganography involves the imperceptible embedding of secret messages within images for covert communication. Current steganographic research utilizes the distortion minimization framework [9]. Within this paradigm, the key challenge is to design embedding costs for each embedding unit (pixels or DCT coefficients) within an image. Subsequently, coding schemes like Syndrome-Trellis Codes (STC) [8] and Steganographic Polar Codes (SPC) [20] are then applied to embed secret information, aiming to approach the rate-distortion boundary. In contrast to other information hiding techniques such as watermarking, security is the foremost metric of evaluation in steganography, defined by the method's effectiveness in avoiding detection by steganalyzers.

Embedding costs can be divided into symmetric and asymmetric types. Symmetric steganography operates under the assumption that +1 or -1 modifications on the same embedding unit have the same impact on steganographic security, whereas asymmetric methods recognize different impacts. Among existing steganographic approaches, the majority of embedding costs are symmetric, including both traditional methods manually designed, such as WOW [13], S-UNIWARD [14] and HILL [17], and those based on deep learning techniques, like ASDL-GAN [32], UT-GAN [36], SPAR-RL [30] and steg-GMAN [15]. Current research [18, 24] indicates that symmetric steganography often fails to adequately account for the statistical relationships between neighboring pixels in images, leading to a security enhancement bottleneck. Consequently, the development of asymmetric embedding costs has emerged as a pivotal research direction for enhancing the security of steganography.

Defining or learning an effective asymmetric embedding cost from scratch is relatively challenging, which is why most current efforts start with a given symmetric embedding cost and then obtain an asymmetric one through adjustments. For instance, CMD [18] and Synch [6] advocate for synchronizing modification directions during the adjustment of embedding costs, which markedly enhance the security over their symmetric counterparts. However, these strategies [18, 6, 34, 39, 33] are largely based on heuristic principles that require a deep understanding of image features. In contrast, inspired by the concept of adversarial examples [12], some studies have adopted a learning approach to derive asymmetric embedding costs by countering CNN-based steganalyzers. These methods [4, 19, 22, 23, 26, 27, 29, 31] typically involve pre-training a CNN-based steganalyzer using existing symmetric methods like S-UNIWARD or HILL. Following this, they adjust embedding costs based on the signs and magnitudes of gradients provided by the steganalyzer, adhering to specific modification rules. For instance, the UGS [23] initially identifies embedding units necessitating adjustment by evaluating gradient magnitudes and initial embedding costs, using two manually set ratios. Additionally, a pre-determined parameter is needed to dictate the magnitude of modifications for

each embedding unit. Although these adjusted methods can effectively enhance the corresponding symmetric steganography, they heavily depend on manually defined parameters and rules for adjustment. The selection of these parameters significantly influences their performance. Unlike methods that rely on heuristic principles and adversarial examples, ReLOAD [25] utilizes a network to learn adjustment policies by minimizing the residual distance between cover and stego images. While ReLOAD markedly improves the security performance of current symmetric approaches. However, like UGS, ReLOAD also requires meticulous choice of adjustment amplitude, which significantly influences the security performance. Moreover, the constant amplitude restricts the capacity to fully leverage the potential of the adjustment, given that the optimal amplitude might vary across various regions of an image.

GANs [11] are potent generative models extensively used in various applications, including image generation, editing, inpainting, and style transformation. Unlike these applications, the distinct requirements of steganography, such as ensuring the complete extraction of secret information and making modifications visually undetectable, limit the adoption of GAN-based techniques in image steganography. Consequently, there are only a few steganographic efforts, such as [32, 36, 30, 15, 16, 21], that are designed using GANs. These approaches exhibit superior security compared to traditional methods, highlighting the potent learning capability of GANs in steganography. To our best knowledge, however, there is no existing research that specifically investigates the use of GANs to automatically refine symmetric embedding costs, aimed at enhancing current steganography methods.

This paper presents an innovative GAN-based framework designed to enhance the security of existing symmetric steganography through the automatic adjustment of embedding costs. Initially, in our framework's Initial Embedding Process stage, we calculate the embedding costs for a cover image subjected to symmetric steganography. We then divide this cover image into two distinct, non-overlapping equal sub-images randomly. The first sub-image is employed to embed half of the secret information, leveraging the initial symmetric embedding costs. The embedding costs for the second sub-image are automatically adjusted using the proposed GAN framework. By adopting our phased embedding strategy, we integrate two steganalyzers within the discriminator. These steganalyzers are designed to distinguish the generated stego image from both the original cover image and the partially embedded stego image, thus offering varying guidance to the generator. In addition, we introduce a straightforward yet efficient update strategy to balance the performance of the generator and discriminator as much as possible for ensuring stable and effective training. Overall, the key contributions of this work are highlighted as follows:

- We introduce a GAN-based model capable of autonomously adjusting existing symmetric embedding costs for enhancing steganographic security. This approach significantly differs from previous related steganography approaches that overly relies on manually set adjustment parameters or rules.
- Unlike current GAN-based steganography approaches, our approach harnesses the advantages of our phased embedding technique, enabling the integration of two steganalyzers with varied discrimination tasks within the discriminator.

- In addition, we offer a straightforward update strategy to achieve and maintain a relative stable and effective process during the GAN training stage.
- Through thorough comparative experiments, we show that our approach substantially improves the security of symmetric steganography, setting a new benchmark for state-of-the-art performance. Additionally, we provide detailed ablation studies to validate the effectiveness of our method.

## 2 PROPOSED METHOD

As depicted in Figure 1, the framework of the proposed method consists of two main stages: the Initial Embedding Process and the GAN Framework for Refinement.

In Stage #1, we commence by calculating the preliminary embedding probability map, $P_1$, for the input cover image $X$ utilizing a prior symmetry method, $S_{init}(\cdot)$, like HILL. Given that the cover image $X$ is defined as $X = [x(i, j)]^{H \times W}$ with dimensions $H \times W$, and the probability map $P_1$ as $P_1 = [p_1(i, j)]^{H \times W}$, where $p_1(i, j) \in [0, 1]$ indicates the likelihood of the pixel value $x(i, j)$ being altered in data embedding process. Concurrently, $X$ is randomly divided into two sub-images with equal size, identified by $Mask_1 = [mask_1(i, j)]^{H \times W} \in \{0, 1\}$ and $Mask_2 = [mask_2(i, j)]^{H \times W} \in \{0, 1\}$, where $Mask_1$ assigning a value of 1 when the elements are located at the first sub-image and assigning 0 when the elements are located at the second sub-image, and inversely for the $Mask_2$. Subsequently, half of the secret information is embedded into the first sub-image marked by $Mask_1$ using STC, resulting in a modification map $M_1 = [m_1(i, j)]^{H \times W}$, where $m_1(i, j) \in \{-1, 0, +1\}$, and the corresponding partially embedded stego image $Y_1$.

In Stage #2, a GAN is designed to modify the embedding costs on the remaining sub-image and complete the embedding of the remaining secret information. In the subsequent sub-sections, we will individually and thoroughly detail the design aspects of the generator, discriminator, and update strategy in Stage #2 within the GAN framework.

### 2.1 Generator

**Design of Structure:** In the proposed method, the generator receives two inputs: the initial symmetric embedding probability map $P_1$ and a modification map $M_1$, which result from the Initial Embedding Process. Leveraging the alterations indicated by $M_1$, the generator aims to adjust $P_1$ in the second sub-image (identified by $Mask_2$) to achieve a more secure asymmetric embedding probability. Subsequently, it embeds the remaining secret information into this modified probability landscape.

To this end, we firstly employ an adjustment network for the generator, illustrated in Figure 2. This network resembles the U-Net architecture, comprising an encoder and decoder layer. The encoder layer consists of 8 down-sampling blocks, while the decoder layer comprises two branches with 8 up-sampling blocks each. Every block consists of two convolution layers, followed by batch normalization and LeakyReLU activation. Unlike previous GAN-based steganographic methods such as [32, 36, 30], which typically employ the common U-Net architecture, taking the cover image as input and outputting the corresponding symmetric embedding probabilities, the adjustment network takes the concatenation of

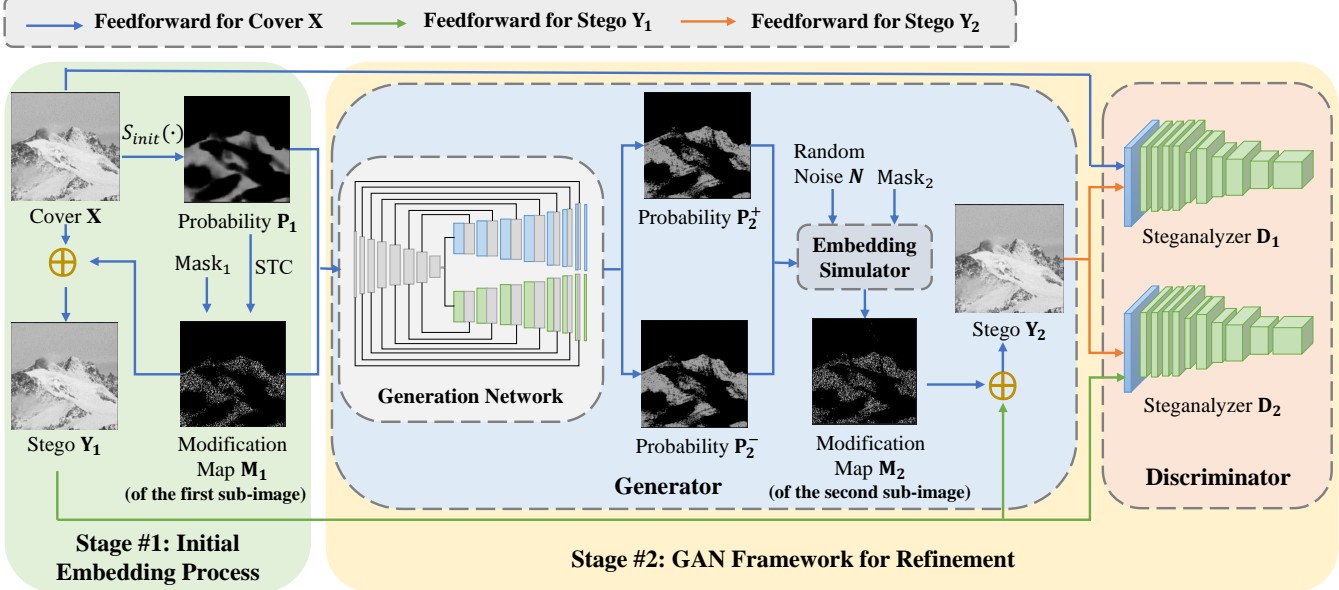

**Figure 1: The framework of the proposed method.**

$M_1$ and $P_1$ as input, and directly outputs the adjusted asymmetric probability map $P_2^+ = [p_2^+(i, j)]^{H \times W}$ and $P_2^- = [p_2^-(i, j)]^{H \times W}$. In these maps, $p_2^+(i, j)$ and $p_2^-(i, j)$ represents the probability of being embedded +1 and -1 for the pixel value $x(i, j)$ of the cover image $X$. It should be noted that, during the Initial Embedding Process, the sub-image within the cover image $X$, identified by $Mask_1$, has been utilized to embed half of the secret information. Therefore, while the probability maps $P_2^+$ and $P_2^-$ cover the entire image, attention will be focused solely on the probabilities associated with the second sub-image. This strategy is employed to streamline the embedding process for the remaining secret information.

Considering that the STC embedding process initially involves converting embedding probabilities into embedding costs (as indicated in Equation (2)), followed by the execution of STC on the computed embedding costs, this procedure is non-differentiable and notably slow. To circumvent these limitations, an embedding simulator is employed during the training of the GAN model. This simulator uses modified embedding probabilities ($P_2^+$ and $P_2^-$) to effectively simulate the information embedding process. Specifically, we firstly generate a random noise matrix $N = [n(i, j)]^{H \times W}$, where each element $n(i, j)$ falls within the range $[0, 1]$. Following this, the modification map $M_2$ is derived by comparing the noise element with the corresponding embedding probability that are located at $Mask_2$, as detailed below:

$$m_2(i, j) = \begin{cases} +1, & n(i, j) < p_2^+(i, j) \ \& \ mask_2(i, j) = 1, \\ -1, & n(i, j) > 1 - p_2^-(i, j) \ \& \ mask_2(i, j) = 1, \\ 0, & otherwise. \end{cases} \quad (1)$$

Then we add the partially embedded stego image $Y_1$ and the resulting modification map $M_2$ to achieve the final stego image $Y_2$ which contains the whole secret information. This process ensures that half of the secret information is embedded within the sub-image

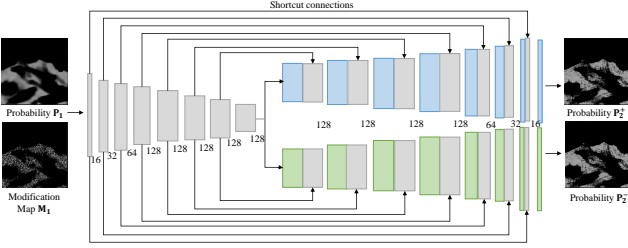

**Figure 2: The structure of the adjustment network in the proposed method.**

identified by $Mask_1$, and the remaining half is embedded within the sub-image identified by $Mask_2$.

It should be emphasized that the embedding simulator is deployed exclusively during the GAN model's training phase. Upon completing the training, the embedding costs $\rho_2$ are calculated using the following equations:

$$\begin{cases} \rho_2^+ = ln(1/p_2^+ - 2), \\ \rho_2^- = ln(1/p_2^- - 2), \\ \rho_2^0 = 0. \end{cases} \quad (2)$$

Utilizing these costs, we can generate the actual stego images through the STC process.

**Design of Loss Functions:** To guide the generator's updates, the corresponding loss function is defined as follows:

$$l_G = \alpha \cdot l_G^1 + \beta \cdot l_G^2 + \gamma \cdot l_G^3, \quad (3)$$

comprising the adversarial loss $l_G^1$, the entropy loss $l_G^2$ and the reconstruction loss $l_G^3$. Here $\alpha$, $\beta$ and $\gamma$ are the weights to attain

a balance between the amplitudes of the loss components. In our experiments, we set $\alpha = 1$, $\beta = 10^{-6}$ and $\gamma = 10$. Next, we will discuss these three distinct losses in detail.

**Adversarial Loss:** The adversarial loss $l_G^1$ aims at guiding a secure modification against discriminator, which is given by:

$$l_G^1 = -\frac{1}{H \times W} \sum_{\forall(i,j)} r(i,j) \cdot [log(p_2^+(i,j)) \cdot \delta(m_2(i,j) = +1) + \qquad (4)$$
$$log(p_2^-(i,j)) \cdot \delta(m_2(i,j) = -1)],$$

where $p_2^+(i,j)$, $p_2^-(i,j)$ and $m_2(i,j)$ respectively denote the $(i,j)$-th element in the probability map $P_2^+$, $P_2^-$ and the modification map $M_2$. $\delta(\cdot)$ represents an indicator function. The weight $r_{i,j}$ for position $(i,j)$ is defined as follows:

$$r(i,j) = \epsilon \cdot m_2(i,j) \cdot g(i,j) \cdot t(i,j), \qquad (5)$$

where $\epsilon = 10^7$. The gradient component $g(i,j)$ represents the gradient at position $(i,j)$ provided by the discriminator, to be discussed in Equation (10) in the subsection 2.3. This encourages modification in the direction of the gradient, as when the sign of $m_2(i,j)$ matches that of $g(i,j)$, it increases $r(i,j)$, aiming at confounding the discriminator's ability to distinguish. The residual component $t(i,j)$ is the $(i,j)$-th element of $T$, which is the absolute values of $Y_1$'s residual filtered by a $3 \times 3$ Laplacian filter kernel with a center element 8, aimed at concentrating modifications in regions characterized by complex textures.

**Entropy Loss:** The entropy loss $l_G^2$ is to ensure the embedding capacity within the second sub-image, and is defined as follows:

$$l_G^2 = (c - H \times W \times q/2)^2, \qquad (6)$$

where $q$ denotes the embedding payload and $c$ is the capacity computed based on the embedding probabilities within the second sub-image:

$$c = -\sum_{\forall(i,j)} \sum_{\forall k} p_2^k(i,j) \cdot log(p_2^k(i,j)) \cdot mask_2(i,j), \qquad (7)$$

where $k \in \{-1, 0, +1\}$. Due to the second sub-image embedding only half the amount of secret information, the target embedding capacity in the loss function (6) is also halved.

**Reconstruction Loss:** As outlined earlier, the proposed adjustment network alters embedding probabilities across the whole image, not just within the area defined by $Mask_2$. Given that the sub-image indicated by $Mask_1$ already contains embedded data, our goal is to maintain the probabilities in the first sub-image and limit adjustments solely to the second sub-image. To achieve this, the reconstruction loss $l_G^3$ is specifically engineered to maintain the embedding probabilities within the first sub-image, identified by $Mask_1$. This mechanism is encapsulated in the following expression:

$$l_G^3 = \frac{1}{H \times W} \sum_{\forall(i,j)} \sum_{\forall k} |(p_2^k(i,j) - p_1^k(i,j)) \cdot mask_1(i,j)|, \qquad (8)$$

where $k \in \{-1, 0, +1\}$ and $p_1^k(i,j)$ represents the initial embedding probability to embed $k$ for pixel $x(i,j)$. Due to the symmetric nature of the initial embedding probabilities, $p_1^+(i,j)$ and $p_1^-(i,j)$ are both equal to $p_1(i,j)/2$, and $p_1^0(i,j) = 1 - p_1(i,j)$.

It's important to highlight that this particular loss function is not found in earlier GAN-based steganography methods. Our design is tailored to the unique requirements of our adjustment task, taking

cues from the domain of image inpainting [38, 28, 40]. In these works, reconstruction loss is frequently used to ensure the inpainted image closely matches the original in areas that are not modified, thereby maintaining the integrity of the unaltered sections while effectively restoring the altered or missing areas. This principle mirrors the objectives of our task. Furthermore, empirical evidence from our experiments suggests that incorporating reconstruction loss positively influences the generator's performance, enhancing it to a certain degree, notably achieving over an 1% improvement against the steganalyzer Yedroudj-Net [37].

## 2.2 Discriminator

**Design of Structure:** The primary goal of the discriminator is to distinguish between the input cover image $X$ and the stego image $Y_2$ that contains all the secret information, created by the generator. In our proposed framework, the process of embedding information is divided into two phases: initially embedding half of the secret information onto the sub-image identified by $Mark_1$, and subsequently embedding the remaining secret information onto another sub-image identified by $Mark_2$. To fully leverage the information from the stego images generated during these two phases to guide the modification of embedding probabilities, we incorporate two steganalyzers, $D_1$ and $D_2$, into the discriminator, each tasked with specific discriminating functions.

Specifically, besides setting up steganalyzer $D_1$ to differentiate the cover image $X$ from the final stego image $Y_2$, we recognize that $Y_2$ is modified on the basis of the stego $Y_1$. Therefore, the modification locations and their directions in $Y_2$ (i.e., $M_2$) are influenced by the first round of steganographic modifications in $Y_1$ (i.e., $M_1$). Previous steganographic studies such as [6, 18] have indicated that concentrating modifications in specific locations and maintaining consistent modification directions among adjacent pixels can significantly improve steganographic security. Thus, we additionally configure steganalyzer $D_2$ to distinguish between the partially embedded stego image $Y_1$ and the final stego $Y_2$, with the aim of effectively learning the impact of modifications of neighboring pixels between $M_1$ and $M_2$ on steganographic security, and guiding the adjustments to the embedding probabilities accordingly. By integrating feedback from both steganalyzers, the discriminator provides diverse guidance to the generator.

It should be noted that, unlike existing GAN-based steganographic methods that rely on a single discriminator, as referenced in [32, 36, 30], and those utilizing multiple discriminators, such as [15, 16], we propose a steganographic framework based on two discriminators with two distinct discriminating tasks. In contrast, in current GAN steganography approaches, the task of the discriminator is uniform, focusing solely on distinguishing between the cover and the final obtained stego image.

**Design of Loss Functions:** To update the discriminator, we take the cross-entropy loss as the discriminant loss as follows:

$$l_{D_1} = -[z_0 log(D_1(X)) + z_1 log(D_1(Y_2))]$$
$$l_{D_2} = -[z_0 log(D_2(Y_1)) + z_1 log(D_2(Y_2))], \qquad (9)$$

where $D_1(X)$ and $D_1(Y_2)$ indicate the output classification vectors for the cover image $X$ and the final stego image $Y_2$ by $D_1$. Likewise, $D_2(Y_1)$ and $D_2(Y_2)$ denote the output classification vectors for the

partially embedded stego images $Y_1$ and $Y_2$ by $D_2$. The ground truth label for $X$ and $Y_1$ is denoted as $z_0$, while for $Y_2$ it is denoted as $z_1$.

## 2.3 Update Strategy

For stable training in GANs, maintaining a balance between the generator and the discriminator is essential. Our proposed method, which incorporates two steganalyzers $D_1$ and $D_2$ with distinct discriminating tasks into the discriminator, further complicates this balance, as the generator must effectively deceive both steganalyzers simultaneously.

Due to the differences in embedding capacity modifications, it is evident that $D_1$'s task of distinguishing between the cover image $X$ and the final stego image $Y_2$ is relatively simpler than that of $D_2$, which distinguishes between the partially embedded stego image $Y_1$ and the final stego image $Y_2$. Hence, $D_1$ is relatively more adept at discerning the output of the generator compared to $D_2$. Efforts should be made to narrow the performance gap between the generator and $D_1$. Furthermore, considering the crucial role of $D_1$ in the security evaluation for steganography, prioritizing its guidance is essential. Taking all these factors into account, we propose an update strategy to strive to ensure balance between the generator and each steganalyzer in the discriminator, as well as ensuring that each steganalyzer operates according to its respective importance level, thereby enabling a stable and effective training process.

The proposed update strategy consists of two aspects. On the one hand, we decrease the initial learning rate of $D_1$ to weaken its discriminatory capability, thereby reducing the performance gap between $D_1$ and the generator. On the other hand, we increase the weight of $D_1$'s gradient feedback to the generator compared to that of $D_2$, as depicted in Equation (10). This enhances the generator's focus on $D_1$ and further bridges the gap between them. Additionally, it amplifies the guidance provided by $D_1$ compared to $D_2$, assigning it a more significant role. The gradient component $g(i, j)$, as previously presented in Equation (5), comprises two parts: the partial derivative of the loss function $l_{D_1}$ and $l_{D_2}$ with respect to the modification $m_2(i, j)$, respectively, which is given by:

$$g(i, j) = \eta \cdot \frac{\partial l_{D_1}}{\partial m_2(i, j)} + \frac{\partial l_{D_2}}{\partial m_2(i, j)}. \tag{10}$$

$\eta$ is a parameter to control the weight of the gradients from $D_1$. We set $\eta = 5$ based on experiments. Further details regarding the update strategy will be provided in Section 3.4.

## 3 EXPERIMENTS

### 3.1 Experimental Setups

In our experiments, we firstly pre-trained our proposed GAN model with 40,000 images sourced from SZUBase [32]. Following this, we employed the pre-trained generator to adjust the embedding costs for a specific steganography (i.e., WOW [13], S-UNIWARD [14], HILL [17], and steg-GMAN [15]). Leveraging the adjusted embedding costs, we then produced stego images through the application of STC on an additional compilation of 10,000 images drawn from both BOSSBase [3] and BOWS2 [2]. This process enabled us to create a dataset of 10,000 cover-stego image pairs. For the evaluation phase, we randomly selected 5,000 of these pairs for the training set, reserved 1,000 pairs for validation, and utilized the remaining

4,000 pairs for the final evaluation. Following the precedent set by ReLOAD, we standardized the resolution of all source images to $256 \times 256$ pixels using MATLAB's imresize function, ensuring uniformity across our datasets. Our experiments considered two different embedding payloads, 0.2 bpp and 0.4 bpp. To demonstrate the efficacy of our approach, we conducted comparative analyses with three established steganography techniques known for embedding costs adjustment: CMD [18], UGS [23], and ReLOAD [25]. Moreover, the security are evaluated using five steganalyzers, including one traditional method SRM [10] and four CNN-based methods, namely, Xu-Net [35], Yedroudj-Net [37], SRNet [5], and Deng-Net [7].

In our model training process, we configured the batch size to 24. The initial learning rates were set at $10^{-4}$ for both the generator and steganalyzer $D_2$, and $10^{-5}$ for steganalyzer $D_1$. These rates are programmed to reduce to 40% of their previous values every 20 epochs. Our training regimen spanned 72 epochs in total. For optimization, we employed the Adam optimizer, configured with beta values of 0.5 and 0.999. To support reproducibility and further research, the source code of our model will be made available online upon the paper's acceptance.

### 3.2 Comparison with Related Methods

In this section, we begin by comparing the security performance of the proposed method to that of three related methods based on embedding costs adjustment: CMD [18], UGS [23], and ReLOAD [25]. The comparative results are shown in Table 1. From the table, we can formulate the following conclusions:

- First of all, the proposed method surpasses the three compared methods in terms of overall performance. Remarkably, it consistently achieves either the first or second highest security performance, underscoring our method's effectiveness. For instance, when considering S-UNIWARD at 0.4 bits per pixel (bpp), our method registers the highest detection error rates across all five steganalyzers. It shows improvements of 1.03%, 0.73%, 5.43%, 2.20%, and 3.15% respectively, when compared to the second-best performing methods. Such enhancements are considerable within the domain of image steganography.
- Compared to the four baseline methods, which include three conventional approaches (WOW, S-UNIWARD, and HILL) and a current leading symmetric embedding cost model based on deep learning (namely, steg-GMAN), our proposed method consistently enhances performance across all cases. For example, in comparison with steg-GMAN, our method shows performance improvements of 2.75% against SRM, 4.71% against Xu-Net, 3.93% against Yedroudj-Net, 4.37% against SRNet, and 5.56% against Deng-Net, respectively, at the payload of 0.4 bpp. However, the method CMD even fails to surpass the performance of steg-GMAN when evaluated against CNN-based steganalysis models.

We also compare the training time and the average time to generate stego images. For a fair comparison, all experiments are conducted on the same server, equipped with an Intel(R) Core(TM) i7-6900K CPU @ 3.20GHz and 4 NVIDIA TITAN X GPUs. The results are presented in Table 2. From this table, we observe that the training time for our method is 13.87 hours, which is slightly slower

**Table 1: Detection error rate (%) of the proposed method and related methods. In the following tables, values with an asterisk(*) denote the best performance in the corresponding case, while the values with an underline denote the second best. The values in parentheses represent differences compared to the baseline, with blue indicating improvement and red indicating decline.**

| Steganalyzer | Method | WOW [13] | | S-UNWARD [14] | | HILL [17] | | Steg-GMAN [15] | |
|---|---|---|---|---|---|---|---|---|---|
| | | 0.2 bpp | 0.4 bpp | 0.2 bpp | 0.4 bpp | 0.2 bpp | 0.4 bpp | 0.2 bpp | 0.4 bpp |
| SRM | Baseline | 36.03 | 24.65 | 36.79 | 24.26 | 41.65 | 30.10 | 42.69 | 33.81 |
| | CMD [18] | 38.78(2.75) | 28.38(3.73) | 39.94(3.15) | 29.56(5.30) | 43.38(1.73) | 35.77*(5.67) | 43.24(0.55) | 35.92(2.11) |
| | UGS [23] | 38.70(2.67) | 29.08(4.43) | 38.32(1.53) | 27.06(2.80) | 42.95(1.30) | 33.68(3.58) | 44.30*(1.61) | 35.71(1.90) |
| | ReLOAD [25] | 37.32(1.29) | 27.12(2.47) | 38.20(1.41) | 25.99(1.73) | 42.96(1.31) | 33.20(3.10) | 43.56(0.87) | 36.08(2.27) |
| | Proposed | 39.35*(3.32) | 29.18*(4.53) | 41.30*(4.51) | 30.59*(6.33) | 43.50*(1.85) | 34.63(4.53) | 44.02*(1.33) | 36.56*(2.75) |
| Xu-Net | Baseline | 36.39 | 24.10 | 40.12 | 28.41 | 39.55 | 30.42 | 42.40 | 36.28 |
| | CMD [18] | 40.96(4.57) | 30.96(6.86) | 44.14(4.02) | 35.19(6.78) | 44.81(5.26) | 37.49(7.07) | 43.45(1.05) | 32.94(−3.34) |
| | UGS [23] | 40.30(3.91) | 29.90(5.80) | 41.93(1.81) | 31.73(3.32) | 43.51(3.96) | 34.99(4.57) | 45.16(2.76) | 37.86(1.58) |
| | ReLOAD [25] | 42.17(5.78) | 31.40(7.30) | 43.04(2.92) | 31.54(3.13) | 44.35(4.80) | 36.03(5.61) | 45.93*(3.53) | 39.79(3.51) |
| | Proposed | 42.75*(6.36) | 34.20*(10.10) | 44.60*(4.48) | 35.92*(7.51) | 45.11*(5.56) | 38.84*(8.42) | 45.80(3.40) | 40.99*(4.71) |
| Yedroudj-Net | Baseline | 23.61 | 14.04 | 31.23 | 17.64 | 29.79 | 19.96 | 41.93 | 33.07 |
| | CMD [18] | 26.10(2.49) | 16.83(2.79) | 35.51(4.28) | 23.56(5.92) | 36.00(6.21) | 26.71(6.75) | 40.20(−1.73) | 31.21(−1.86) |
| | UGS [23] | 32.75(9.14) | 21.12(7.08) | 36.41(5.18) | 22.74(5.10) | 39.30(9.51) | 28.23(8.27) | 43.35(1.42) | 35.28(2.21) |
| | ReLOAD [25] | 25.28(1.67) | 16.00(1.96) | 32.53(1.30) | 20.60(2.96) | 34.91(5.12) | 25.20(5.24) | 43.60(1.67) | 36.58(3.51) |
| | Proposed | 33.13*(9.52) | 23.53*(9.49) | 38.99*(7.76) | 28.99*(11.35) | 41.59*(11.80) | 33.14*(13.18) | 43.96*(2.03) | 37.00*(3.93) |
| SRNet | Baseline | 22.06 | 12.25 | 25.14 | 13.89 | 29.44 | 19.91 | 35.69 | 27.84 |
| | CMD [18] | 23.70(1.64) | 14.48(2.23) | 28.16(3.02) | 17.33(3.44) | 32.57(3.13) | 23.06(3.15) | 36.45(0.76) | 26.04(−1.80) |
| | UGS [23] | 28.97*(6.91) | 17.50(5.25) | 30.51(5.37) | 17.65(3.76) | 35.55*(6.11) | 24.72(4.81) | 39.54(3.85) | 32.40*(4.56) |
| | ReLOAD [25] | 22.71(0.65) | 13.78(1.53) | 27.78(2.64) | 16.34(2.45) | 30.75(1.31) | 20.98(1.07) | 38.45(2.76) | 30.21(2.37) |
| | Proposed | 28.44(6.38) | 18.47*(6.22) | 31.55*(6.41) | 19.85*(5.96) | 34.24(4.80) | 25.68*(5.77) | 40.26*(4.57) | 32.21(4.37) |
| Deng-Net | Baseline | 20.81 | 11.34 | 25.32 | 12.25 | 27.54 | 18.65 | 36.68 | 27.09 |
| | CMD [18] | 22.49(1.68) | 12.86(1.52) | 28.15(2.83) | 16.24(3.99) | 32.51(4.97) | 21.54(2.89) | 34.17(−2.51) | 27.00(−0.09) |
| | UGS [23] | 28.05*(7.24) | 17.05(5.71) | 30.19(4.87) | 16.65(4.40) | 33.77(6.23) | 22.02(3.37) | 36.64(−0.04) | 29.90(2.81) |
| | ReLOAD [25] | 22.25(1.44) | 12.41(1.07) | 27.21(1.89) | 14.04(1.79) | 30.56(3.02) | 20.09(1.44) | 38.62(1.94) | 31.89(4.80) |
| | Proposed | 27.65(6.84) | 17.19*(5.85) | 32.13*(6.81) | 19.80*(7.55) | 35.06*(7.52) | 25.48*(6.83) | 41.83*(5.15) | 32.65*(5.56) |

**Table 2: The time for training (hour) and generating a stego image (second) of the proposed and related methods. Note that CMD does not require training.**

| Methods | CMD | UGS | ReLOAD | Proposed |
|---|---|---|---|---|
| Training (h) | / | 9.78* | 85.49 | 13.87 |
| Generating a stego (s) | 0.05 | 2.04 | 2.66 | 0.04* |

than the 9.78 hours required by UGS, yet significantly shorter than the 85.49 hours required by ReLOAD. Once training is complete, our proposed method requires only 0.04 seconds to generate a stego image, a duration similar to that required by CMD, and considerably less than the times required by UGS and ReLOAD.

## 3.3 Comparative Study on Different Steganalyzers in Discriminator

In the proposed model, we incorporate two steganalyzers (i.e., $D_1$ and $D_2$) within the discriminator, each tasked with unique classification roles. Precisely, their objective is to distinguish the generator's output (i.e., the stego image $Y_2$) from the cover image $X$, and the stego image $Y_1$ that has undergone partial information embedding. In this section, we undertake an experiment involving various combinations of four typical steganalyzers: Xu-Net, Yedroudj-Net, SRNet, and Deng-Net for $D_1$ and $D_2$. For the sake of simplicity, we limit our scenarios to those where the same steganalyzer is utilized

for both $D_1$ and $D_2$, and the original embedding cost is calculated by the steganography HILL at 0.4 bpp. The results are summarized in Table 3. From this table, it is evident that:

- Using two steganalyzers typically results in improved security compared to relying on just one. In our setup, we utilized Deng-Net as both $D_1$ and $D_2$, achieving the highest level of security in most scenarios. Employing a single steganalyzer, particularly $D_2$, proved to be less effective. This highlights the benefit of combining two steganalyzers, each analyzing distinct inputs, to provide comprehensive insights, substantially boosting the robustness of steganographic security.
- The selection of steganalyzers significantly influences the security effectiveness of the proposed method, whether employing a single steganalyzer or a combination of two. Specifically, utilizing Deng-Net as $D_1$ in a single steganalyzer setup, or as both $D_1$ and $D_2$ in a dual steganalyzer configuration, demonstrates a noticeable advantage.

## 3.4 Comparative Study on Different Update Strategies

In the proposed model, we allocate two steganalyzers, namely $D_1$ and $D_2$, in the discriminator and implement a straightforward update strategy to maintain equilibrium between the generator and discriminator for stable training. This is achieved by decreasing the initial learning rate of $D_1$ and enhancing the weight $\eta$ of its gradient feedback to the generator. In this section, we compare our approach

**Table 3: Detection error rate (%) of the proposed method with different combinations of steganalyzers in Discriminator.**

| Combinations | Configuration | SRM | Xu-Net | Yedroudj-Net | SRNet | Deng-Net | Average |
|---|---|---|---|---|---|---|---|
| Single Steganalyzer $D_1$ | Xu-Net | 33.70 | **40.64**$^*$ | 28.35 | 24.42 | 20.96 | 29.61 |
| | Yedroudj-Net | 33.81 | 38.79 | 32.76 | 24.31 | 23.10 | 30.55 |
| | SRNet | 33.66 | 37.49 | 28.94 | 22.58 | 18.96 | 28.33 |
| | Deng-Net | 34.19 | 38.46 | 33.18 | 25.15 | 24.81 | 31.16 |
| Single Steganalyzer $D_2$ | Xu-Net | 34.27 | 37.36 | 26.59 | 22.76 | 21.71 | 28.54 |
| | Yedroudj-Net | 34.06 | 36.22 | 27.59 | 24.89 | 22.60 | 29.07 |
| | SRNet | 33.93 | 36.92 | 26.80 | 23.29 | 22.16 | 28.62 |
| | Deng-Net | 33.81 | 36.14 | 26.99 | 24.59 | 23.14 | 28.93 |
| Two Steganalyzers, $D_1 = D_2$ | Xu-Net | 34.56 | 39.71 | 28.80 | 24.11 | 21.07 | 29.65 |
| | Yedroudj-Net | 33.22 | 38.61 | **33.49**$^*$ | 23.84 | 22.91 | 30.41 |
| | SRNet | 34.28 | 38.05 | 32.45 | 25.49 | 24.34 | 30.92 |
| | Deng-Net (proposed) | **34.63**$^*$ | 38.84 | 33.14 | **25.68**$^*$ | **25.48**$^*$ | **31.55**$^*$ |

**Table 4: Detection error rate (%) of the proposed method with different update strategies or parameters.**

| Strategies | | SRM | Yedroudj-Net | Deng-Net | Average |
|---|---|---|---|---|---|
| Strategy #1 | | 34.04 | 28.77 | 24.54 | 29.11 |
| Strategy #2 | | 34.60 | 29.73 | 24.58 | 29.64 |
| Strategy #3 | | 34.11 | 29.47 | 24.48 | 29.35 |
| Strategy #4 | | 33.52 | 27.16 | 22.78 | 27.82 |
| Proposed $\eta =$ | $*$ | **34.75**$^*$ | 31.25 | 24.59 | 30.20 |
| | 0.5 | 33.78 | 28.24 | 23.96 | 28.66 |
| | 1 | 34.57 | 29.10 | 24.30 | 29.32 |
| | 5 | 34.63 | **33.14**$^*$ | **25.48**$^*$ | **31.08**$^*$ |

**Table 5: Detection error rate (%) of the proposed method with different inputs to the Generator.**

| Inputs | SRM | Yedroudj-Net | Deng-Net | Average |
|---|---|---|---|---|
| Input #1 | 30.20 | 26.34 | 22.33 | 26.29 |
| Input #2 | 34.15 | 32.81 | 24.39 | 30.45 |
| Input #3 | 34.20 | 33.13 | 25.44 | 30.92 |
| Proposed | **34.63**$^*$ | **33.14**$^*$ | **25.48**$^*$ | **31.08**$^*$ |

with four alternative strategies as outlined below. Besides, we also investigate on different parameters $\eta$ in the proposed strategy. Note that $\eta = *$ indicates that $\eta$ is adaptively learned from the generator, while $\eta = 0.5$ signifies that the generator is instructed to focus more on $D_2$ instead of on $D_1$.

- **Strategy #1:** The generator is updated with both $D_1$ and $D_2$, with the initial learning rate and gradient weight of $D_1$ set identical to those of $D_2$. This implies that $D_1$ and $D_2$ are directly combined without any modulation.
- **Strategy #2:** The generator is updated with only $D_2$ for the initial 1/2 training period (i.e., 30 epochs) and then combines $D_1$ and $D_2$ for the remaining epochs.
- **Strategy #3:** The generator is updated with both $D_1$ and $D_2$, but $D_1$ is updated every 5 epochs, which means the update times of $D_1$ is 1/5 of that of the generator.
- **Strategy #4:** Both $D_1$ and $D_2$ share the same parameters, indicating using a single steganalyzer to differentiate between two distinct classification tasks.

The comparative results are shown in Table 4. From Table 4, we observe that the choice of update strategy plays a crucial role in enhancing the security performance of our proposed method. Specifically, by reducing the initial learning rate and fixing $\eta$ at 5, our strategy achieves superior performance compared to Strategies #1 to #4, with an impressive average performance boost of 31.08%, an improvement of at least 1.44%. This represents a commendable

advancement in the field of image steganography security. Secondly, the application of the weight $\eta$ within our model also impacts its security effectiveness. Compared to a learnable $\eta$ (i.e., $\eta = *$), using a large $\eta$ can achieve better security performance. However, if $\eta \leq 1$, it would significantly reduce security.

### 3.5 Comparative Study on Different Inputs to the Generator

In the proposed method, we concatenate the modification map $M_1$ and the initial embedding probability map $P_1$ as the inputs of the generator, as illustrated in Figure 2. In this section, three alternative forms of input to the generator are included for comparative study, which are designed as follows:

- **Input #1:** We only input $P_1$ to the generator, excluding $M_1$, in order to explore the effects of $M_1$ on adjustments.
- **Input #2:** We input the concatenation of $P_1$ and the residual of $M_1$, extracted by a $3 \times 3$ filter with 0 at the middle and 1 around, which tends to capture neighboring modifications instead of directly feeding $M_1$ to the generator, following the principle of typical steganography CMD [18].
- **Input #3:** We employ a generator with two encoding branches and input $P_1$ and $M_1$ respectively to one branch. This aims at investigating on the impact of employing multiple branches instead of multi-channel inputs.

The comparative results are presented in Table 5. From the table, we observe three key points: 1) The modification map $M_1$ is highly beneficial for the generator in adjusting to obtain safe embedding costs. This is evident as the results of Input #1 demonstrate a significant decrease of 4.43%, 6.80%, and 3.15%, respectively, when compared

**Table 6: Detection error rate (%) of the proposed method with different partition proportions for the two sub-images.**

| $Mask_1$-$Mask_2$ | SRM | Yedroudj-Net | Deng-Net | Average |
|---|---|---|---|---|
| 30% - 70% | 34.33 | 31.84 | **26.55*** | 30.91 |
| 40% - 60% | 34.36 | **33.19*** | 25.62 | 31.05 |
| 50% - 50% | 34.63 | 33.14 | 25.48 | **31.08*** |
| 60% - 40% | **34.76*** | 30.55 | 23.94 | 29.75 |
| 70% - 30% | 32.96 | 27.28 | 21.04 | 27.09 |

**Table 7: Detection error rate (%) and training time (hour) of the proposed method with varying numbers of sub-images used in image partitioning.**

| # of sub-images | 1(HILL) | 2(Proposed) | 3 | 4 |
|---|---|---|---|---|
| SRM (%) | 30.10 | 34.63 | 35.72 | **36.39*** |
| Yedroudj-Net (%) | 19.96 | 33.14 | 34.62 | **36.02*** |
| Deng-Net (%) | 18.65 | 25.48 | 26.84 | **27.42*** |
| Training Time (h) | / | **13.87 *** | 30.71 | 42.63 |

to the proposed method; 2) After first performing filtering preprocessing on the modification map $M_1$ and then inputting it along with the probability map $P_1$ into the generator, it does not enhance the steganographic security performance; 3) Incorporating a dual encoding branch for $M_1$ and $P_1$ in the generator does not contribute to enhancing security performance and instead increases model complexity. Although this approach performs better than other inputs, it still falls short of the effectiveness achieved by directly concatenating $M_1$ and $P_1$ for the input.

## 3.6 Comparative Study on Different Partition Proportions

In the proposed method, we divide the secret messages into two equal halves for embedding into each sub-image with equal size. In this section, we explore scenarios where the cover image is partitioned into varying proportions, with secret information allocated accordingly. This includes proportions of 30%-70%, 40%-60%, 50%-50% (proposed), 60%-40%, and 70%-30%. The results are summarized in Table 6. From Table 6, we obtain two following observations:

- On average, equal partitioning achieves the highest security performance. As the difference in the partitioning ratio increases—for example, changing from an equal partition to more imbalanced ratios like 30%-70% or 70%-30%—a significant decrease in security performance is observed.
- For the steganalyzer Deng-Net, allocating a larger proportion of the secret message to the second sub-image (indicated by $Mask_2$) enhances security performance. The primary reason is that Deng-Net serves as the targeted steganalyzer ($D_1 = D_2 =$ Deng-Net) within the proposed discriminator.

## 3.7 Comparative Study on Different Number of Sub-images

In our prior methodological discussions, we began by randomly splitting the cover image into two non-overlapping sub-images.

Initially, we utilized an existing symmetric steganography method to embed half of the secret information into one of the sub-image. Then, we employed a trained GAN model to adjust the embedding costs for the other sub-image, facilitating the embedding of the remaining secret information. Importantly, our approach can be expanded to divide the image into multiple sub-images, with each sequentially embedding a corresponding portion of information until the entire secret message is concealed.

In this section, we investigate the effects of varying the number of sub-images (1, 2, 3, 4) on the steganographic security and the efficiency of the training process. Here, the use of 1 signifies no division into sub-images, depending entirely on the original HILL steganography for data embedding. The comparative results are detailed in Table 7. It is observed that with an increase in the number of divisions, there is a progressive improvement in the security of image steganography. Specifically, when the division increases to four, our method shows an approximate 2% performance enhancement compared to the earlier experiments with two divisions. However, this improvement in performance comes with an increase in training time. Since the adjustment and embedding of each sub-image require the full image information for GAN training, each stage of GAN training for one sub-image takes approximately 14 hours, leading to an overall training time that increases linearly with the number of divisions. Considering the training time, we only conducts experiments with the image divided into two sub-images previously.

## 4 CONCLUSION

In this paper, we introduced an innovative GAN-based framework to enhance the security of existing symmetric steganography methods by adjusting their embedding costs to asymmetric ones. Our approach overcomes the limitation of previous related techniques that heavily relied on heuristics and manually defined parameters or rules. Tailoring to the unique demands of steganographic tasks, we meticulously designed a steganographic framework utilizing GAN, encompassing generators, discriminators, and their respective loss functions. Notably, in line with our phased embedding strategy, we introduce a dual-discriminator and dual-tasks mechanism and an associated update policy, ensuring a stable and efficacious training regime. Through extensive comparative analysis, our approach is shown to outperform traditional symmetric steganography techniques and other relevant methods that aim to adjust embedding costs, marking a significant advancement in the field.

While our framework represents a significant step in the research on automatic embedding costs adjustment, certain aspects warrant further investigation. To better capture the correlation and interaction of neighboring pixel modifications, we plan to explore an improved architecture for the generator, potentially incorporating an attention module or drawing inspiration from network architectures used in image inpainting tasks. Additionally, we will explore adaptively introducing noise to the discriminator's input as a data augmentation technique, contingent upon its relative performance compared to the generator, thereby achieving dynamic balance between the two. Furthermore, given the scarcity of methods addressing embedding costs adjustment for JPEG images, we intend to extend our framework to the JPEG domain in future research.

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
