# OpenReview forum: "GAN-based Symmetric Embedding Costs Adjustment for Enhancing Image Steganographic Security"
_acmmm.org/ACMMM/2024/Conference — MM2024 Poster_

### Official Review · Reviewer_n4dh · 2024-05-20

**Rating:** 4
**Confidence:** 2

**Summary:**

The paper proposes an asymmetric embedding cost for adaptive image steganography, which uses a GAN to adjust the symmetric embedding cost generated by the previous method, Hill. Specifically, the authors first embed secret data into half of the cover image using the symmetric embedding cost to obtain a partially embedded stego image. Then, they adjust the symmetric embedding cost using a GAN to obtain the asymmetric embedding cost. Finally, they embed the secret data into the partially embedded stego image to obtain the final stego image. Two discriminators are used to distinguish between the cover, partially embedded stego, and stego images.

**Strengths:**

- The paper is well-structured and easy to follow.

- The design of discriminator is interesting.

- The experimental results look promising.

**Limitations:**

- The key of adaptive steganography is to design embedding cost. Despite different methods following different technical paths, the final evaluation ultimately depends on the quality of the embedding distortion. Why not compare the proposed method with the GAN-based techniques [15, 16, 21].

- The amplitude significantly influences the performance of the compared method.  How is the magnitude in the comparison methods selected? Is it set to be the same, or based on the default settings of the original work?

**Suitability:**

3

---

### Official Review · Reviewer_NcFR · 2024-05-23

**Rating:** 5
**Confidence:** 4

**Summary:**

This paper introduces a novel embedding costs adjustment method based on Generative Adversarial Networks (GAN), aimed at addressing the limitations associated with prior methods that heavily rely on handcrafted parameters or predefined rules for adjustment. A GAN-based framework is used to automatically transform symmetric costs into asymmetric costs without manual intervention, and a two-stage embedding process is designed to realize the embedding of secret messages. To accommodate the two-stage embedding process, the discriminator is combined by two steganalyzers. The experimental results show that the proposed approach significantly improves security performance compared to related methods, and achieves the state-of-the-art results.

**Strengths:**

1. The approach proposed in this paper is novel. The authors propose to use GAN to automatically optimize the symmetric embedding cost without manual intervention in related methods. In addition, this paper further improves the security performance by integrating two steganalyzers for different tasks respectively in the discriminator.

2. This paper has clear logic and sufficient experimental verification. The authors clearly illustrate the specific scheme proposed in this paper through charts and detailed formulas. Meanwhile, the authors design experiments from multiple perspectives to verify the effectiveness of the proposed method.

**Limitations:**

1. Some of the descriptions are not precise enough. Adjustment methods can generally be categorized into additive and non-additive approaches. For example, CMD is a non-additive method, whereas ReLOAD is an additive method. Therefore, it is crucial to clearly state in the introduction that the proposed method adopts a non-additive approach. In conclusion section, the mention of a "dual-discriminator" appears contradictory to the previous statement, which indicated the incorporation of two steganalyzers into one discriminator. Consistency in terminology should be ensured throughout the paper.

2. Some of the content is not explained in sufficient depth. In the proposed method, the cover image is randomly divided instead of in the way of CMD or Synch. What are the specific advantages of random division? In addition, Section 2.2 elucidates the rationale behind incorporating steganalyzer D_2, aiming to learning the impact of modification between M_1 and M_2 by differentiating between Y_1 and Y_2. However, a more detailed explanation of the connection is needed to improve clarity.

3. Lack of description of the learning rate setting for the discriminator. In Section 2.3 the authors propose to balance the performance difference between the two classification tasks by reducing the initial learning rate of D_1, but the paper does not explain how to reduce the learning rate of D_1. Considering that Section 3.4 shows that the updating strategy of the discriminator has a large impact on the experimental results, the initial settings of the learning rates of D_1 and D_2 should be stated and the impact of the different settings should be explored.

**Suitability:**

3

---

### Official Review · Reviewer_mY1v · 2024-05-25

**Rating:** 4
**Confidence:** 4

**Summary:**

In this paper, the authors propose a GAN-based framework to enhance the security of existing symmetric steganography methods by adjusting the embedding costs to asymmetric ones. The proposed method addresses the limitation of previous works which rely on heuristics and manually defined parameters or rules. Experimental results show that the proposed method outperforms several SOTAs of symmetric steganography in terms of detection error rate against steganalyzers.

**Strengths:**

-The two-stage steganography framework is somewhat novel, which includes an initial embedding process and a GAN-based refinement.
-The authors incorporate two steganalyzers into the discriminator, each of which is customized for specific tasks. The results in Table 3 do support their claim.
-Comparison against other methods has been included diligently.

**Limitations:**

-The motivation of randomly dividing the cover image into two non-overlapping sub-images of the same size should be clarified.
-Explanation of why certain choices were made such as the 8 up-sampling blocks in adjustment network, the weights corresponding to three losses in Equation 3 and the weight of gradient from D1 in Equation 10.
-More details of the adjustment network should be provided in Figure 2.
-Clarify the training process of the five steganalyzers including SRM [10], Xu-Net [35], Yedroudj-Net [37], SRNet [5], and Deng-Net.
-The authors only evaluate the performance in terms of detection error rate. Other metrics will also be helpful for discussion.

**Suitability:**

2

---

### Official Review · Reviewer_TYbF · 2024-05-25

**Rating:** 3
**Confidence:** 4

**Summary:**

This paper introduces an advanced GAN-based framework that transitions symmetric costs to asymmetric ones without the need for the manual intervention seen in existing approaches, to improve the security of steganography. The framework starts by creating symmetric costs for a cover image split into two sub-images, with part of the secret information embedded in one. A GAN model is then used to adjust the costs of the second sub-image asymmetrically to securely embed the rest of the information. The GAN's discriminator uses two steganalysis for diverse guidance, and a stable training update strategy is implemented.

**Strengths:**

1、The paper introduces a novel application of GANs in steganography, automatically adjusting embedding costs without relying on manually defined parameters.

2、The introduction of a stable training process and a simple yet effective update strategy ensures the model's robustness and reliability.

3、Comprehensive experiments demonstrate significant security improvements over existing symmetric steganography methods.

**Limitations:**

1、This article does not illustrate related work. It is very difficult for readers to understand the improvements they have made and to measure innovation.

2、For steganography methods, the PSNR of original container image and decoded image, show the effect of steganography. This result is also not given. The authors not only need to show that their method has better security but also need to show advantages in terms of fidelity and robustness.

3、The author does not provide any subjective comparisons, which makes it difficult for readers to intuitively feel the advantages of the method.

**Suitability:**

3

---

### Meta-Review · Area_Chair_drdp · 2024-07-03

**Recommendation:** Accept (Poster)
**Confidence:** 4

**Metareview:**

One weakly accept, two borderline accepts and one accept.